# Performing Group-Based Physical Activity (Gbpa) in the Work-Place: Survey and Sociological Considerations of the "Happy Bones" Project

Francesca Romana Lenzi [1], Eliana Tranchita [1], Elisa Grazioli [1], Claudia Cerulli [1,*], Vincenzo Esposito [2], Giuseppe Coppola [1], Elisa Moretti [1], Caterina Mauri [1], Carlo Minganti [1] and Attilio Parisi [1]

[1] Department of Movement, Human and Health Sciences, University of Rome Foro Italico, 00135 Rome, Italy
[2] Department SARAS, Department of History, Anthropology, Religions, Arts and Entertainment, Sapienza University of Rome, 00185 Rome, Italy
* Correspondence: claudia.cerulli@uniroma4.it; Tel.: +39-063-6733-303

**Abstract:** The goal of the following work was to identify the effects, positive or negative, of performing group-based physical activity (GBPA) in the workplace. In addition, the scope of the present research was to investigate some social and relational aspects of medical origin associated with the Happy Bones project. The sample consisted of 28 women between 47 and 67 years old, employees of the University of Rome "Foro Italico", in menopause, and inactive. The explorative nature of the investigation and the multidimensional aspect of the variables suggested the adoption of a qualitative method. Even though the survey did not fulfil the minimum standards of representativeness, interview analysis showed a positive trend in joining physical activity in the workplace, as shown by the good compliance of the participants with the proposed workplace training protocol. Personal motivation linked to the project itself or to the corresponding activity existed albeit to a secondary extent; the unifying element of the group existed regardless of the project and was due to the home institution, hence to the workplace.

**Keywords:** physical activity; qualitative methods; osteoporosis; prevention; compliance

## 1. Introduction

Many studies have shown the importance of the social but also relational aspects of physical activity (PA) [1,2], a topic that has considerable relevance in the work environment, with important implications for healthcare. A fundamental premise of the research presented is to emphasize the sustainability dimension concerning the preventive nature of the Happy Bones project. The concept of prevention is closely related to sustainability; prevention means controlling as far as possible the consequences of a risk [3–5] and thus acting to reduce both the economic costs of treatment and the human costs of illness. The European project Happy Bones "Physical Activity in women in menopause: a collaborative partnership for active lifestyles for the prevention and treatment of osteoporosis" has been funded with support from the European Commission (G.A. 613137-EPP-1-2019-1-IT-SPO-SCP–ERASMUS+ SPORT). Coordinated by Università di Roma "Foro Italico", the partnership consists of one Italian and four European partners: Istituto per lo Sviluppo Socio-Economico–ISES (IT) (Partners); Bulgarian Sports Development Association–BSDA (BG) (Partners); Alexandru Ioan Cuza University of Iaşi–UAIC (RO) (Partners); Gazi University–GU (TURKEY) (Partners); and Fundaciò Salut I Envelliment UAB-FSIE (SPAIN) (Partners). Postmenopausal women are the population at the highest risk of developing osteoporosis and suffering fractures. In fact, the peak bone mass reached in the mid-twenties remains relatively stable until the onset of menopause, which involves a rapid period of bone loss [6,7]. The proportion of women at high fracture risk who did not receive treatment was 71% in 2019 (up from 59% in 2010). In the scientific literature, it is now clear



that PA and nutrition represent the two non-pharmacological key points for the prevention and treatment of osteoporosis. Numerous recent studies seem to indicate that the most effective training model should aim to be combined (aerobic + strength training), for at least 8 months, with a frequency of at least 3 days a week, and should integrate exercises for balance and mobility [8–11]. Despite all these recommendations, the population in general, and women in particular, do not reach the minimum amount of PA recommended. As it is known, the workplace represents an environment in which individuals spend a high number of hours and covers a wide age group with various risk factors, and it is often the work itself that leads individuals to high levels of sedentary lifestyle, spending many hours sitting [12]. Moreover, the type of work itself could cause damage to workers' health and targeted physical activity (PA) protocols may be able to prevent it [13–15]. Indeed, structured and supervised physical exercise interventions carried out in the workplace, with a minimum duration of four months and with at least two weekly sessions, allow a significant decrease in BMI, an increase in muscle mass [16,17], and an improvement in systolic blood pressure [18]. Consequently, promoting and developing a PA protocol in the workplace can represent a focal point in the prevention, treatment and management of several diseases, including osteoporosis in postmenopausal women. Moreover, it could increase the PA compliance as well as the general health of workers.

In this light, the HB project aims, in the first place, to promote PA for menopausal women through an innovative physical exercise protocol, aimed at the psycho-physical well-being of the participants and to increase participation and awareness of the importance of PA. Moreover, the project evaluates compliance through the number of attendances of each participant at 72 supervised lessons provided for by the HB protocol. Compliance is expressed as a percentage of the overall lessons [19,20]. The possible dropout of participants and the timing of the dropouts were recorded. The adoption of therapies aimed at a sustainable approach involves the assessment of the risk of the disease, but also of the lack of adherence to the preventive pathway for non-medical reasons. In this case, we talk about social determinants [21,22] of health that affect the effectiveness of health strategies in both prevention and treatment. The present work reports the preliminary results of an HB protocol performed in Italy.

The work presented in this paper follows a line of research called medical sociology. This branch of sociology today is an academic and university reality. In Italy, it is one of the research lines of the Ais (Association of Italian sociology) Sociology of Health and Medicine section. Medical sociology is the sociological analysis of medical organizations and institutions, knowledge production and selection methods, actions and interactions of health care professionals, and the social or cultural (rather than clinical or bodily) effects of medical practice. In medicine, "social explanations" of disease etiology have meant for some physicians a reorientation of medical thinking away from purely clinical criteria of the disease. The introduction of "social" factors into medical explanations has manifested itself most strongly in the branches of medicine closely linked to the community: Social medicine and, later, general medicine [23,24].

This work investigated group dynamics in the execution of a physical training protocol aimed at postmenopausal women with high risk of developing osteoporosis.

Here, following the example of other research carried out with a sample with different characteristics [25,26], it was decided to investigate how the activity carried out in a group affects the degree of participation in the PA performed and whether it affects it positively or negatively. Finally, the work also aimed to investigate the social determinants of health [27–29]. In line with the perspective adopted here, the research framework involved the use of qualitative [30] or non-standard methods [31–33]. The cognitive objective presented hitherto requires this research method, since only this methodological approach is potentially adequate for making explicit what the subjects think about their reference reality (representations, values, attitudes, explicit and tacit knowledge, intuitions, etc.).

## 2. Materials and Methods

Thirty female subjects were recruited by the University of Rome "Foro Italico" through an institutional mail sent by the human resources office. Inclusion criteria were being an employee of the "University of Rome Foro Italico", aged between 47 and 67 years, inactive, and being in post-menopause (Figure 1).

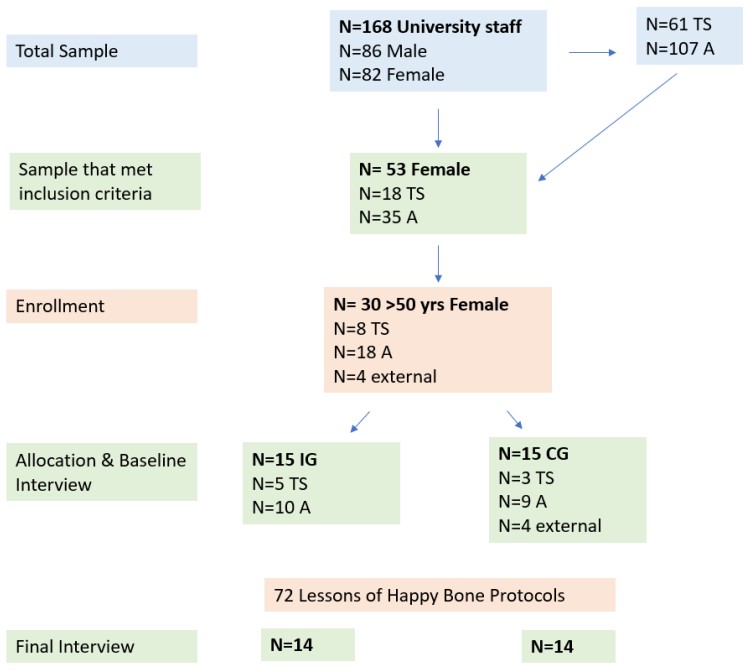

**Figure 1.** Flow-chart of the study.

The proposed study protocol was drafted in accordance with the European Union's Standards of Good Clinical Practice and the current revision of the Declaration of Helsinki and was approved by the University Ethics Committee. The participants, after signing the informed consent, were allocated, according to their preferences, in the intervention group (IG *n* = 15), which performed 6 months tailored combined protocol "Happy Bones", or control group (CG *n* = 15), which continued with the daily routine. The first group was made up of employees at the University of Rome "Foro Italico", both as technical-administrative staff and as teaching staff, while the second group also included four external women that met the inclusion criteria, necessary to reach the number of 15 participants per group. Those of them with cardiovascular disease and other complications were excluded (Table 1).

**Table 1.** Patients' characteristics. Abbreviation: I—intervention group; CG—Control Group; BMI—body mass index.

| Patients' Characteristics | IG *n* = 15 (M ± SD) | CG *n* = 15 (M ± SD) | IG + CG *n* = 30 (M ± SD) |
|---|---|---|---|
| Age (yrs) | 52 ± 5.66 | 57.87 ± 3.31 | 57.24 ± 4.47 |
| Weight (Kg) | 54.5 ± 2.12 | 65.02 ± 9.31 | 65.15 ± 9.35 |
| Height (cm) | 163 ± 1.41 | 161.46 ± 5.33 | 163.1 ± 5.23 |
| BMI | 24.14 ± 3.18 | 24.60 ± 3.55 | 25.8 ± 3.325 |

In this case, the method selected was the unstructured interview [34,35]. The use of a tool belonging to the family of non-directive techniques is due to the fact that they are the most suitable for collecting narratives and life stories because they allow the respondents

to express themselves with forms of communication that are most appropriate [34,36–38]. Interviews were conducted by three members of the research team. In this way, the interview protocol was refined for gathering the information needed to answer our research objective. The interviews were carried out and recorded for the most part live in compliance with the COVID-19 regulations in force; the respondents who could not participate in the live interview were heard through video chat channels such as Teams and Zoom.

The interview protocol conducted consisted of six very general open-ended questions, with which the interviewees were able to express their opinions and considerations regarding the preventive exercise protocol carried out with Happy Bones. The first two questions, which were used by the interviewers to build trust and put the respondents at ease, were aimed at surveying the respondents' opinions about today's society's relationship with their bodies and with physical activity (PA). Thus an attempt was made to identify, in the wake of the work of Mauss (1936) [39] and Loudcher (2011) [40], how the view and use in sports of the body has changed over the years. The next three questions were constructed with the aim of surveying respondents' views on the performance of preventive PA carried out in groups [1,2]. The last question was administered only to participants in the intervention group and was intended to detect and evaluate [41] the preventive activity performed. Given the purpose of this paper, the analysis of opinions collected from the analysis of questions three, four and five is presented in this article. The interviews were conducted at two different points in time because the work presented was structured with the specifics of a longitudinal study in mind [42]. Visser [43] asserts that a longitudinal study involves examining the changes that occur over time in the same sample of subjects. In this way, the processes of change directly associated with the passage of time or with the presence of a phenomenon occurring in the course of the time elapsed between the different phases of the research are studied. The initial interviews were submitted in the first weeks of the project (between October and November 2021), while the second round of interviews were conducted in the final weeks of the project (March/April 2022).

"HAPPY BONES" TRAINING PROTOCOL.

The HB training protocol, developed by two researchers in the field of preventive and adapted PA, is a complex and innovative training protocol that aims to prevent and reduce bone loss, especially in the lumbar spine and femoral neck, in postmenopausal women [10]. The program includes home training (5 days a week) and supervised training (3 days a week) performed in university facilities, which includes group training, strength training, and cardiovascular training, as explained below (Table 2).

**Table 2.** Structure of the Happy Bone training protocol. Abbreviation: Mon—Monday; Wed—Wednesday; Fri—Friday.

| HOME TRAINING (5 Days/Week) | SUPERVISED TRAINING (3 Days/Week) |
| --- | --- |
| Single-leg standing exercise<br>Star-Excursion Balance exercise | Session 1 (Mon) Group training, Cardiovascular training, and Strength training<br>Session 2 (Wed) Cardiovascular training and Strength training<br>Session 3 (Fri) Group training, Cardiovascular training, and Strength training |

HOME TRAINING.

Participants are required to perform two exercises at home, the "Single leg Standing" and the "Star-Excursion Balance" [44].

SUPERVISED TRAINING.

The supervised training is composed of three different phases: (1) group training; (2) strength training; (3) cardiovascular training, starting with a warm-up and ending with a cooldown of the major muscle groups involved in the protocol. The strength training and the cardiovascular training must be performed 3 days per week, while the group training only in 2 of the 3 days. (Table 2).

The second group, the control group, was instead made up of women, both employees of the "Foro Italico" and external to the institution, who had in common the fact that they did not take part in the adapted PA protocol (hereinafter referred to as the "control group").

The participation of the respondents was voluntary, and each participant had the right and opportunity to withdraw from the research at any time, even without specifying the reasons for their choice. The analysis was conducted in a totally anonymous way as it was carried out by eliminating any reference that could allow the statements to be connected to a specific person.

## 3. Results

Initially, the sample consisted of 30 respondents, however, since one member of the control group and one member of the intervention group for personal reasons could not conduct the second interview, the final sample of the sociological analysis results was 28. Following the thematic analysis to analyze the interview data, responses were categorized on the basis of three categorical dimensions:

1. Consent granted to the performance of group PA;
2. Consent granted to the performance of PA in the workplace;
3. Attribution of an added value in terms of physical or psychological well-being and commitment to the purpose (Control Group and Intervention Group).

The opinions of the respondents of both groups on these topics were collected and analyzed. It should be noted that the transcription of the interviews and therefore the excerpts reported here are as faithful as possible to the spoken words to avoid the risk of interpretative distortions. The affiliations to the working sector (teacher–PD, technical-administrative staff–PTA, laboratory technician–TL) are indicated in order to detect, as explained below, similarities of answers based on belonging to a working sector.

### 3.1. Group-Based Physical Activity (GBPA)

Within the interviews conducted, a part was aimed to sample the social dimension of sport, namely whether the sporting activity carried out in the workplace in a group is more productive. Specifically, the Happy Bones project involves individuals in group-based PA in the workplace. Therefore, within this field, it was also sampled whether, according to the interviewees, such group-based PA might be considered a way to tighten new friendships or create new ones. In this case, a premise is a must; most of the interviewees built their own path, first of training and then of career, within the University of Rome "Foro Italico". This is a privileged place to play sports and, as the interviewees said several times, it is the place where some of them had grown up together and had worked together for more than 30 years:

> "*no, in a sense I repeat that perhaps from the fact that in any case it has been 30 years, 35 years since I have known them and therefore it is obvious that in my opinion there was not even this aspect to deal with simply because you know that by going there you know everyone*" (PTA, 62 years).

Two main positions were formed within the group. The first position identifies also a social and socialization moment in sports practice:

> "*doing sport is better together with other people, i.e., I like it more with friends because yes... because you become a group and beyond doubt, it improves the friendship relationship*" (PD, 56 years).

This position is delved into more deeply by the group that carried out the activity of Happy Bones; during the second survey, after having carried out the activity, the group indicated that the moment in which you do a PA together is a moment in which you know each other better and in greater depth between colleagues from different sectors and professional fields:

*"There is always a way to improve relations with colleagues, perhaps by seeing them in a different context you also discover other things, knowing them in other ways... my colleagues, people whom I might know less but I was seeing in the university, of whom I might not have known anything, absolutely nothing. And that is a moment of sharing, beyond the sport but also beyond other aspects because in any case, while maybe you are doing a treadmill you say things, in a word, you compare yourself, it is also an important moment for this"* (PTA, 55 years).

A big part of the control group also shared the opinion that doing activities in a group creates new friendships and has a very high social value. Finally, those who supported the position favorable to the practice of group-based PA indicated the sociability derived from PA was a motivational stimulus to ensure consistency in the performance of the activity itself:

*"I think, instead that on the contrary it is an extra motivation, because socialization is a fundamental aspect of PA, you know very well especially, when we talk about the third age I do not say in my case but well... (smiles)... I'm getting there, but let's say that it is important to have the spirit of support of your colleagues"* (PD, 59 years).

The second, minority position, among the opinions of the interviewees, held that group-based PA is not a moment of socialization and that practicing activities together with colleagues does not bring particular benefits. The main reason for this position was the need to separate your work and private life:

*"yes, in my opinion, there is a need to separate, I do not know why but this is the impression... you may even have lunch, or an aperitif as soon as you finish work, but in reality, when you get off you want to hang out with friends and therefore friends are not colleagues, you must also separate the roles"* (retired, 62 years).

Differences between the surveys:

Also with regard to this question, the interviewees from both groups maintained their positions over time. The group that participated in the Happy Bones (IG), a group-based PA with colleagues, agreed that doing PA in a group is useful for strengthening or creating new friendships, both during the first and second interviews, and having chosen to carry out the Happy Bones activity could have reinforced that opinion [45]. In the control group, there are more people who held the opinion that PA in a group does not help to socialize, often also expressing their own personal choice of wanting to exercise alone. In the first and second surveys, the members of the control group also confirmed their opinion without changing their points of view.

*3.2. PA in the Workplace*

The subject matter that will be analyzed in this section deals with the topic of PA practiced in the workplace. The focal point of the survey was the sensations and benefits generated by practicing activities without having to move from the workplace and aimed to detect the perception of motor activity in the workplace by the interviewees.

All the interviewees expressed a positive opinion about the reference target, underlining its various strong points:

*"Well, for me it can only bring advantages, absolutely. I believe that also in other European countries in many enterprises... There is precisely the promotion of PA inside the companies themselves just like welfare, as a package to make their employees feel good; therefore, I think it is absolutely a pleasant thing"* (PTA, 56 years).

In this excerpt, a common thread emerged from the interviewees which underscored the positive aspects of PA in the workplace, underlining the importance of promoting PA within companies, stimulating it in various European realities, considered as examples to be followed in terms of employee welfare and well-being. These considerations were upheld, both between the first and second interviews, and between the two different groups analyzed, with particular pieces of evidence for the advantages prompted by the

elimination of travel times to reach the place where to do PA and, in the case of the "Foro Italico", from the sports setting already connected to the workplace. The only exception in terms of logistics involved with the PA in the workplace was represented by one interviewee who, in addition to underlining the advantages of the work environment and the zero commuting to the place where to do PA, highlighted the lack of home comfort after carrying out activities in equipped centers:

> *"The advantages are those of great comfort, by reducing to zero the commuting time from wherever one is to the place where to do exercises. Logistically, here it is more uncomfortable because to go back to the office, you have to change, give yourself a quick wash that is less pleasant than maybe relaxing at home"* (PD, 61 years).

In the following excerpt, the interviewee highlighted the community factor created by the workplace, which generates an impact on the belonging of the people involved and an extra stimulus in relation to the PA carried out in the workplace:

> *"Positive aspects definitely yes, you create this sort of community, there is also a greater sense of belonging to the workplace here"* (PD, 58 years).

Compliance with HB's training protocol was categorized into quartiles, based on participation rate (Figure 2). Five participants were placed in the high compliance group (75–100%); seven participants in the moderate compliance group (50–75%); two participants in the low compliance group (25–50%); and no participants in the very low compliance group (0–25%). Only one of the participants recruited at the beginning of the project dropped out. As can be seen from these data, our intervention produced good results in terms of compliance with the training protocol.

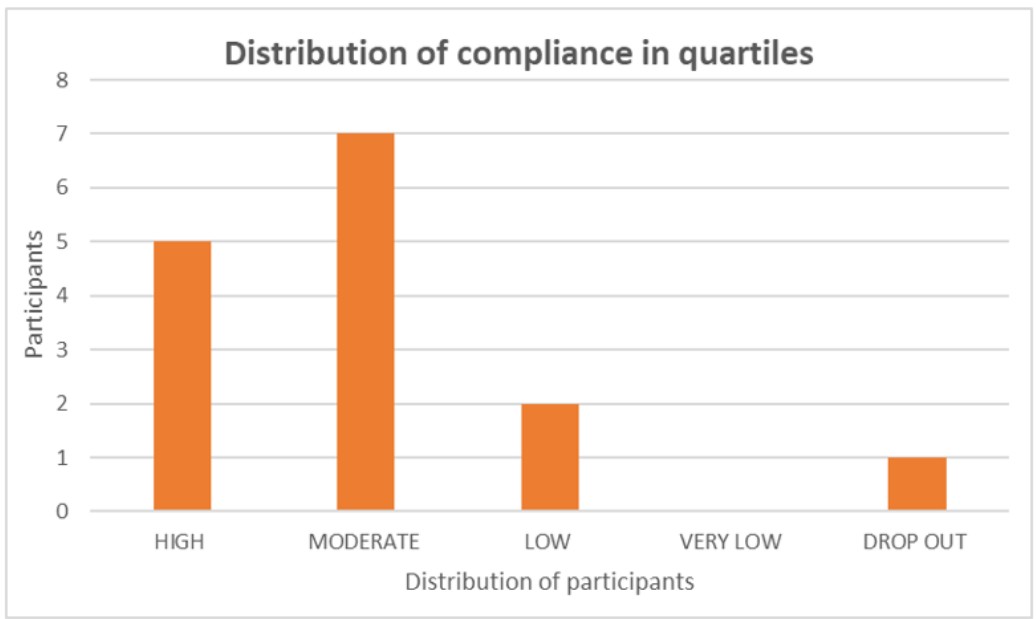

**Figure 2.** Distribution of intervention group participants based on compliance categorized into quartiles. High compliance 75–100%; moderate compliance 50–70%; low compliance 25–50%; very low compliance 0–25%; drop out.

### 3.3. Physical and Psychological Well-Being after Physical Activity (in the Workplace)

The degree of well-being, both from a physical and mental point of view, perceived by the interviewees upon returning to work after having carried out the activities was also a subject of the survey. The goal of this question was to detect whether the interviewees, upon returning to work after having carried out PA, had noticed a greater or lesser level of both physical and psychological well-being. From the control group, the interviewees did not express negative opinions towards the idea of practicing PA in the workplace,

expressing a common line of thinking between the first and second interview regarding both physical and psychological well-being generated by motor activity, specifically emphasizing the optimization of travel times to reach the gym as a fundamental detail to reduce stress and increase feelings of benefit. Within the action group, the common line of thinking concerned the perception of well-being upon returning to work, in greater or lesser quantities depending on the type of activity carried out and one's training background:

> *"Certainly not being too intense and not being too tired I have not noticed, let's say, an important change with respect to the quality of the subsequent work. I noticed an improvement in mental well-being before and after, linked to the idea of the activity"* (PTA, 56 years old).

In this excerpt, the interviewee highlights the type of work performed but emphasizes the importance of perceived well-being both before and after exercising in the workplace. A common feature among the interviewees that generated only one exception, in particular from a physical point of view as described in the following passage:

> *"This thing about the equipment that was increasing the weight like this for me was too tiring. One positive objective that it has, it had it let's say on a psychological level"* (PD, 62 years).

In the following excerpt, the respondent also highlighted the benefits found after exercising about the psychological wellbeing as well as with regard to the physical and spinal problems and pain experienced:

> *"I had some... positive feelings certainly on a psychological level because I like doing physical activity anyway, I also experienced an improvement in some of the aches and pains I may have had coming from my spine"* (PD, 61 years).

Both in the control group and in the action group, the interviewees showed a perception of physical and psychological well-being upon returning to work after having carried out PA. As evidenced also by the enclosed excerpts, the common thread among the two groups linked PA in the workplace to the sensations of benefit experienced before and after working out in the gym.

## 4. Discussion

The study carried out reveals convergences, but also elements of disagreement between the two groups of interviewees and between the two surveys. However, some divergences are affirmed in the analysis with regard to group-based PA. The interviewees who participated in the Happy Bones project and built their professional training path within the University of Rome "Foro Italico", highlighting the importance of belonging to the institution in their career, stressed the usefulness of group-based PA as an important component in strengthening or creating new friendships. In the control group, on the other hand, opinions that disagreed with this point of view emerged; the interviewees expressed a preference for exercising alone. This point reveals a research bias, consisting of the belonging of the intervention group to the institution of "Foro Italico", with a high vocational component and, therefore, with a high impact on identity level [46,47], and a sense of belonging and groupality. Both groups in the two surveys expressed positive opinions regarding PA at the workplace, emphasizing the advantages stemming from the elimination of travel times to reach the place where to do activities. Among the two, the intervention group highlighted the importance of doing PA at the "Foro Italico", underlining the link between one's work environment and the sporting environment, emphasizing the sensations of both physical and psychological well-being perceived after exercising. Social relations, body care, and new positive stimuli to face age-related health risks, however, represented the convergences expressed with regard to the Happy Bones project, which collected positive opinions linked to the objectives of the experience by both groups and in both surveys. Finally, the degree of motivation that led to participation in the activity

should be considered. In this case, it was possible to detect different attitudes that prompted the interviewees to take part in the Happy Bones project.

We have divided the interviewees into two groups based on professional category, technical administrative staff (in the following analysis this category includes both technical administrative staff and laboratory staff, due to the fact that the responses of both groups were perfectly overlapping) and teaching staff. The responses of those who indicated as the main incentive to carry out the Happy Bones activity the fact that other colleagues were participating, and the social aspect of the sport are highlighted in yellow:

*"I have accepted this thing just because it is here, and it is done in a way... i.e., it is personalized, it is in the family so it's not that... I feel much more guided compared to any other gym"* (56 years old).

As can be seen during the first survey, the teaching staff settled on a similar position mostly due to their work and academic background, declaring to have carried out the Happy Bones activity because it was an experimental arrangement developed by their home university. While those who were part of the administrative staff of "Foro Italico" carried out the activity mainly because of personal and/or social well-being.

In this survey, the teaching staff embraced the position of those who carried out the Happy Bones activity because they believed in the experimentation promoted within the "Foro Italico" (in blue in the graph):

*"The stimulating factor for me is to be included in an experimental activity, because I also do many, even in we are in this together and I know what it means the support and not having participants for the conclusions, I felt a different responsibility"* (65 years).

According to the compliance, the results showed good participation in the Happy Bones protocol with a medium-high participation rate for more than a third of the participants, evidencing a higher compliance than similar studies of PA in the workplace [48,49]. In addition, only one participant withdrew from the project before starting the protocol, due to personal impairments. These results, being above the average of studies in the literature, represent extremely encouraging data and suggest that this intervention can represent an important starting point in the development of interventions to promote health and PA in the workplace.

Although the survey does not comply with the minimum standards of representativeness, amounting rather to exploratory research, some evidence emerges from the interviews, which recalls some general theories on group dynamics [46,50,51] and from which we can deduce some characteristics of the groups and of their longevity, including belonging and sharing a set of values. Nevertheless, with regard to the specific research, it is necessary to clarify some distinctive conditions, which make it hardly generalizable. The first is that the activity related to the Happy Bones project is not a sport in itself, but rather an activity dedicated to individuals with specific physiological needs; in other words, it is not strictly speaking a sport, but it is an activity mostly ascribed to rehabilitation or physical health practice. Consequently, it can be defined as an activity allocated more to the sphere of duty than to that of pleasure, more to the sphere of PA for health, and less to that of sports. If the two dimensions may have in common a motivation linked to body care, in this case, the motivational aspect dictated by the physical conditions of the participants and the risks associated with procrastinating this activity are worsened, while the playful element of pure sporting activity is missing, which would otherwise stimulate the interest and involvement of the participants.

Another element to take into consideration is that the small reference sample is subject to one specific condition. The interviewees were not banded together only by the social (women, workers) and physical (age range, menopause, and inactivity) conditions that made them subjects of interest for the project in question. They also shared an aspect linked to belonging to a university like "Foro Italico", dedicated to sport and motor science, where the workforce, including most of the interviewees, share a strong bond with the home institution and with what it represents. In other words, this bond developed not only and

not so much because of the historical and life bond that many of them had with sport and motor activities in general, but also since, for most of them, this place has been a venue for study and growth, even before their career. Therefore, the reference community has a profound effect on the sense of belonging and connection to the place of work and activity, to the shared values and relational reference system, and to the consequent willingness and motivation to participate in the specific project of interest to the research. In addition, many of them have stressed the importance of being involved in a shared journey, stripped of their working role.

Ultimately, we can say that, as other research has also found [1,2], the creation of a group and identification in the group's dynamics lead subjects to complete their therapeutic exercise path better and without dropping out.

## 5. Conclusions

This paper explores a case study related to the application of a preventive strategy concerning the development of osteoporosis in postmenopausal women and notes the incidence, for the effectiveness of such sustainable prevention, of factors related to sociality, self-perception, and motivation. Among the research results, it emerges that grouping and socialization [46,52] within the motor activity carried out by the observed individuals constitute one of the main social determinants and that such preventive activity is effective and sustainable. In conclusion, one last feature that has strengthened even more the support for the project, even in the cases of individuals less "affiliated" in the narration of their own life path, work, and relationship with PA, was the participation in research promoted by colleagues from the "Foro Italico". This motivation emerged with particular strength among the interviewees belonging to the teaching staff category. More generally, the theme of trust in support of the project and in the effort of continuity is not just about personal motivations, in this case, minority ones, but also, and above all, about reasons linked to the group and the project itself. Therefore, participation in the project had as its most powerful element the one of trust in the project and in those who implemented it, as well as in the fact that it was carried out within one's own institution, whose value system is shared and on which the common long-term path depends. The personal motivation linked to the project itself or to the corresponding activity exists albeit to a secondary extent; the unifying element of the group existed regardless of the project and was due to the home institution, hence to the workplace. It affected the support for the project and the trust granted that stimulated the motivation of almost all the interviewees, a fact also confirmed by the low variation in motivation, which was always strong between the first and second survey. In conclusion, the results from this sociological analysis indicate that the PA protocol proposed in the workplace has the merit of increasing participation of postmenopausal women at risk of osteoporosis through group dynamics and group creation, so it is the authors' opinion that widespread use of this activity or similar activities is desirable.

The research presented in this paper was conducted with a very small sample, which did not allow us to do more sophisticated analyses; moreover, the respondents were all from the same work and academic background. A possible future development could be to conduct the same research with a larger sample that contains within it subjects from different academic and work backgrounds, so as to capture the differences between them.

**Author Contributions:** Conceptualization, F.R.L., A.P.; methodology, C.C., E.T.; formal analysis, V.E., C.M. (Carlo Minganti); investigation, C.M. (Caterina Mauri), E.M., E.G.; data curation, G.C.; writing—original draft preparation, C.C., E.T., E.G.; writing—review and editing, F.R.L.; visualization, F.R.L., C.C.; supervision, A.P. All authors have read and agreed to the published version of the manuscript.

**Funding:** This study was Co-funded by the Erasmus+ Programme of the European Union "Happy Bones 613137-EPPI-1-2019-1-IT-SPO-SCP".

**Institutional Review Board Statement:** The study was conducted in accordance with the Declaration of Helsinki and approved by the Institutional Review Board: CAR106/2021.

**Informed Consent Statement:** Informed consent was obtained from all subjects involved in the study.

**Data Availability Statement:** Not applicable.

**Conflicts of Interest:** The authors declare no conflict of interest.

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
