# Peer review of "Performing Group-Based Physical Activity (Gbpa) in the Work-Place: Survey and Sociological Considerations of the “Happy Bones” Project"

_sustainability, doi:10.3390/su15010480_

Round 1

Reviewer 1 Report

This paper is focused on effect of group based physical activity in the work place. The following issues need some address:
1. The literature review is adequate. Several articles published on effects of workplace training programs in various workplaces need to be reviewed for a better overview. Also, it is important not only to describe and explain the issue of problems but also need to link it with the situations that occur during the work. These recently published articles need to be reviewed and cited (https://doi.org/10.1080/10803548.2015.1029290, https://doi.org/10.1080/10803548.2021.2014090, https://doi.org/10.1177/0269215515575747).
2. The size of the sample is good but the sampling must be described much more clearly. How were the respondents chosen? Why was this choice made? How were they approached? In addition, more demographic info about the respondents would be useful. A pictorial view is better for readers.
3. How the questionnaire was developed? Neither justification of various items nor any supporting literature. Please provide proper justification with questionnaire.
4. The results are of some interest but the authors could elaborate further on the practical implications of their findings.

Author Response

REVIEWER 1

  1. The literature review is adequate. Several articles published on effects of workplace training programs in various workplaces need to be reviewed for a better overview. Also, it is important not only to describe and explain the issue of problems but also need to link it with the situations that occur during the work. These recently published articles need to be reviewed and cited (https://doi.org/10.1080/10803548.2015.1029290, https://doi.org/10.1080/10803548.2021.2014090, https://doi.org/10.1177/0269215515575747).

Thank you for the comment, we added the articles about the effects of workplace training programs you indicated and they were made in the reference section (pag.1 line 44 and pag.2 lines 45-60)

  1. The size of the sample is good but the sampling must be described much more clearly. How were the respondents chosen? Why was this choice made? How were they approached? In addition, more demographic info about the respondents would be useful. A pictorial view is better for readers.

Thank you, the respondents were chosen and approached through an institutional mail (pag.3 line 106), and we have chosen this sample due to the inclusion criteria, and because women at this age are in the menopausal period, and they are at high risk to develop osteoporosis. (pag. 3 line 105-108). A pictorial view was inserted on pag 3 line 109.

  1. How the questionnaire was developed? Neither justification of various items nor any supporting literature. Please provide proper justification with questionnaire.

Thank you for the comment, we added the revisions you indicated in page 4 lines 141-153

  1. The results are of some interest but the authors could elaborate further on the practical implications of their findings.

Thank you for the comment, we included some practical implications in the discussion section (pag.11 lines 411-418 and lines 450-452).

Reviewer 2 Report

The authors of the manuscript entitled "Performing group-based physical activity (GBPA) in the work-place. Survey and sociological considerations of the “Happy Bones” project" identified the effects, positive or negative, of performing group-based physical activity (GBPA) in the workplace and investigated some social and relational aspects of medical origin associated with the Happy Bones project. The study design and concept are good. The aim of the study was clearly defined.

However, I have comments and suggestions regarding the results of the study.

 Materials and Methods

Line 93: The sample consists of 28 womenbut table 2 shows 15 vs 15 participants.

Lines 136-138: After having listened and transcribed all the interviews, 2 members belonging to the control group and 1 member of the Happy Bones group were eliminated from the group of respondents because… Please indicate the correct number of participants in the text and in the table.

Line 147: What is the 6th dimension?

Table 2: I suggest changing the word high to height

  In my opinion, the Discussion section is actually a continuation of the Results section. From this point of view, there is no relevant discussion in the manuscript. Personally, I think that the authors have processed a very interesting and valuable issue that can be discussed in a high quality and to a sufficient extent, while they will be able to confront their observations and findings with the results of other authors. This part therefore needs to be supplemented. Ultimately, I cannot assess whether such a study with the mentioned methodological procedures and common statistical methods can be published. The study lacks exact data. Assuming that it is possible to publish the manuscript, it is necessary to expand the Results section with additional statements and findings of the authors, or to add anthropometric, biochemical, health data, etc.

Author Response

REVIEWER 2

  1. Line 93: The sample consists of 28 women…but table 2 shows 15 vs 15 participants.

Thank you for the comment, we added the revisions you indicated in page 5 lines 205-207

  1. Lines 136-138: After having listened and transcribed all the interviews, 2 members belonging to the control group and 1 member of the Happy Bones group were eliminated from the group of respondents because… Please indicate the correct number of participants in the text and in the table.

Thank you, we corrected the correct number of participants in the text and in the table.

  1. Line 147: What is the 6th dimension?

Thank you for the comment, after your review we noticed an error and deleted this statement

  1. Table 2: I suggest changing the word highto height

Thank you, we changed the word.

  1. In my opinion, the Discussion section is actually a continuation of the Results section. From this point of view, there is no relevant discussion in the manuscript. Personally, I think that the authors have processed a very interesting and valuable issue that can be discussed in a high quality and to a sufficient extent, while they will be able to confront their observations and findings with the results of other authors. This part therefore needs to be supplemented.

Thank you for the comment, we implemented the discussion in pag.11 lines 411-418 and lines 450-452.

  1. Ultimately, I cannot assess whether such a study with the mentioned methodological procedures and common statistical methods can be published. The study lacks exact data. Assuming that it is possible to publish the manuscript, it is necessary to expand the Results section with additional statements and findings of the authors, or to add anthropometric, biochemical, health data, etc.

Thank you for the comment, we do not add anthropometric, biochemical, or health data because these do not fit with the aim of the manuscript, which has a sociological imprint not clinical. Finally, in our manuscript it is not possible to identify statistical inference procedures because the approach used is Qualitative and not Quantitative research. Thus, the analysis is based exclusively on thematic analysis procedures, aimed at emphasizing the identification, analysis and interpretation of patterns of meaning (or "themes") within qualitative data.

Reviewer 3 Report

Reviewer’s comments to author

The present study was designed to identify the effects, positive or negative, of performing group-based physical activity (GBPA) in the workplace. In addition, the scope of the present research is to investigate some social and relational aspects of medical origin associated with the Happy Bones project. Researchers here followed a qualitative approach to investigate the above research questions. To my opinion, the present study may represent an important addition to the existing literature. Therefore, I believe it is publishable in its current form with a major revision. Below, I offer both general and specific comments regarding the current manuscript and, where appropriate, I have made suggestions to the authors regarding potential strategies for addressing these tasks.

General Comments:

·         A innovative topic

·         It covers the aims and the scope of this scientific journal

·         It follows a qualitative approach to investigate the research questions

Moreover,

·         It would be easier for readers if you replace the phrase “physical activity” with PA in all the text.

·         At the Introduction, please add a paragraph describing the problem of osteoporosis in postmenopausal women. What has previous research shown regarding the effect of physical activity in the prevention of women’s osteoporosis?

·         Regarding References please correct all of them based on the Sustainability journal (MDPI) format. In general, please follow the format style of the Sustainability journal (MDPI).

Specific comments:

Abstract

·         Page 1, line 25: Please add a dot after the word “workplace”.

·         Page 1, line 30: Please change “physical activity” to “physical activity (PA)” and then use this shortcut (PA) in all the following text.

Introduction

·         Page 1, lines 29-30: Please add some references here.

·         Page 1, line 39: Please change the phrase “our University”. Please name your Institution here.

·         Page 2, line 44, 47, 71: Please change “physical activity” to “PA”

·         Page 2, line 57: Please delete paragraph here. Just continue the phrase in line 56.

·         Page 2, line 67: Please start a new paragraph here regarding the aims of this work “This work…”

Materials and Methods

·         Regarding interviews, please explain here how you fulfill the criterion of trustworthiness.

·         Please explain in depth the steps you followed to analyze the interviews’ data (e.g. thematic analysis)

·         Page 3, line 86: Please delete “(1985)”

·         Page 3, lines 98-99: Please add here the Reference Number of the University Ethics Committee decision.

·         Page 3, Table 1: Please change the font of the letters in Table 1.

·         Page 3, line 124, 141, 142: Please change “physical activity” to “PA”

·         Page 3, Table 2: Please change the font of the letters in Table 2.

·         Page 3, line 136: Please change number 2 with the word two.

·         Page 3, line 137: Please change number 1 with the word one.

·         Page 3, line 140: Please change number 3 with the word three.

Results

·         Page 5, lines 159, 177, 188, 196, 206, 210, 214, 225: Please change “physical activity” to “PA”

·         Page 6, lines 226, 230, 232, 235, 256, 258: Please change “physical activity” to “PA”

·         Page 6, line 242: Please change number 5 with the word Five

·         Page 7, lines 277, 278: Please change “physical activity” to “PA”

·         Pages 5 and 6: In the Sub-section “3.2. Physical activity in the workplace”, please add some more participants’ quotes.

·         Pages 6 and 7: In the Sub-section “3.3 Physical and psychological well-being after physical activity (in the workplace)”, please add some more participants’ quotes.

Discussion

·         Page 7, lines 284, 288, 294, 297: Please change “physical activity” to “PA”

·         Page 7, line 285: Please change the second “who” with another word e.g. “and built their …”

·         Page 8: I’m not sure if all these (in page 8) are Results or not. If they concern the Results, you need to move them to the Section Results. To my opinion, you also need to remove the Figures 2 and 3 (I cannot understand their purpose here).

·         Page 9, line 340: Please change “physical activity” to “PA”

·         At the Discussion section, please add some results from previous studies to support or not the findings of the present study.

Conclusions

·         Page 7, line 371: Please change “physical activity” to “PA”

·         Please add limitations of the present study and make suggestions for future research.

Author Contributions

·         Page 10, row 388: Please add a dot after the word “manuscript”.

Reviewer’s Decision: Accepted with major revision.

Author Response

REVIEWER 3

  1. it would be easier for readers if you replace the phrase “physical activity” with PA in all the text.

Thank you for the comment, we have replaced all the physical activities in PA in the text

2. At the Introduction, please add a paragraph describing the problem of osteoporosis

In postmenopausal women. What has previous research shown regarding the effect of physical activity in the prevention of women’s osteoporosis?

Thank you, we add a paragraph in the introduction (pag.1 and 2 lines 44-60.

3. Regarding References please correct all of them based on the Sustainability journal (MDPI) format. In general, please follow the format style of the Sustainability journal (MDPI). 

Thank you for the comment, we added the revisions you indicated were made in the reference section

  Specific comments: 

Abstract

  • Page 1, line 25: Please add a dot after the word “workplace”.

Thank you, we added a dot after the word “workplace”

  • Page 1, line 30: Please change “physical activity” to “physical activity (PA)” and then use this shortcut (PA) in all the following text.

Thank you, we follow your comment and used PA in the text

Introduction

  • Page 1, lines 29-30: Please add some references here

Thank you, we added the references

  • Page 1, line 39: Please change the phrase “our University”. Please name your Institution here.

Thank you, we change the phrase that you indicate

  • Page 2, line 44, 47, 71: Please change “physical activity” to “PA”

Thank you, we follow your comment and used PA in the text

  • Page 2, line 57: Please delete paragraph here. Just continue the phrase in line 56.

Thank you, we delete the paragraph

  • Page 2, line 67: Please start a new paragraph here regarding the aims of this work “This work…”

Thank you, we start new paragraph

Materials and Methods 

  • Regarding interviews, please explain here how you fulfill the criterion of trustworthiness.

Thank you, we explain criterion of trustworthiness in the page 4 lines 134-137

  • Please explain in depth the steps you followed to analyze the interviews’ data (e.g. thematic analysis) 

Thank you, we have used themaric analysis for analyze data

  • Page 3, line 86: Please delete “(1985)”

Thank you, we delete 1985

  • Page 3, lines 98-99: Please add here the Reference Number of the University Ethics Committee decision.

Thank you, we added the Reference Number of the university ethics commitee

  • Page 3, Table 1: Please change the font of the letters in Table 1.

Thank you, we change the font

  • Page 3, line 124, 141, 142: Please change “physical activity” to “PA”

Thank you, we follow your comment and used PA in the text

  • Page 3, Table 2: Please change the font of the letters in Table 2.

Thank you, we change the font

  • Page 3, line 136: Please change number 2 with the word two.

Thank you, we change number 2 with word two

  • Page 3, line 137: Please change number 1 with the word one.

Thank you, we change number 1 with word one

  • Page 3, line 140: Please change number 3 with the word three.

 Thank you, we change number 3 with word three

Results

  • Page 5, lines 159, 177, 188, 196, 206, 210, 214, 225: Please change “physical activity” to “PA” 

Thank you, we follow your comment and used PA in the text

  • Page 6, lines 226, 230, 232, 235, 256, 258: Please change “physical activity” to “PA” 

Thank you, we follow your comment and used PA in the text

  • Page 6, line 242: Please change number 5 with the word Five

Thank you, we change numer 5 with the word five

  • Page 7, lines 277, 278: Please change “physical activity” to “PA”

Thank you, we follow your comment and used PA in the text

  • Pages 5 and 6: In the Sub-section “3.2. Physical activity in the workplace”, please add some more participants’ quotes.

Thank you, we added participants’ quotes

  • Pages 6 and 7: In the Sub-section “3.3 Physical and psychological well-being after physical activity (in the workplace)”, please add some more participants’ quotes.

Thank you, we added participants’ quotes

 Discussion

  • Page 7, lines 284, 288, 294, 297: Please change “physical activity” to “PA”

Thank you, we follow your comment and used PA in the text

  • Page 7, line 285: Please change the second “who” with another word e.g. “and built their …”

Thank you, we change the second who

  • Page 8: I’m not sure if all these (in page 8) are Results or not. If they concern the Results, you need to move them to the Section Results. To my opinion, you also need to remove the Figures 2 and 3 (I cannot understand their purpose here).

Thank you, we have remove the figures 2 and 3

  • Page 9, line 340: Please change “physical activity” to “PA”

Thank you, we follow your comment and used PA in the text

  • At the Discussion section, please add some results from previous studies to support or not the findings of the present study.

Thank you, we added some result that support present study

Conclusions

  • Page 7, line 371: Please change “physical activity” to “PA”

Thank you, we follow your comment and used PA in the text

  • Please add limitations of the present study and make suggestions for future research. 

 Thank you, we added limitations in the present study

Author Contributions

  • Page 10, row 388: Please add a dot after the word “manuscript”.

Thank you, we added a dot.

Round 2

Reviewer 1 Report

Thanks for making changes.

Author Response

I thank the reviewer for accepting my revisions

Reviewer 2 Report

I thank the authors for accepting my comments and suggestions, the manuscript has been improved and I agree to its publication.

Author Response

(The authors gave the same response as above.)

Reviewer 3 Report

Reviewer’s comments to author

Reviewer’s Decision: Accepted with minor revision.

Please see below:

·         Page 2, lines 59, 64: Please change “physical activity” to “physical activity (PA)” and then use this shortcut (PA) in all the following text.

·         Page 3, line 104: Please replace 30 with the word “Thirty”

·         Page 4, lines 142, 146: Please change “physical activity” to “physical activity (PA)” and then use this shortcut (PA) in all the following text.

·         Page 5, lines 194-198: Please use full alignments in the text.

Author Response

Dear Reviewer, thanks for accepting our corrections.

We have modified the following points as requested:

Page 2, lines 59, 64: We changed “physical activity” to “physical activity (PA)” and then we used this shortcut (PA) in all the following text.

  • Page 3, line 104: we replaced 30 with the word “Thirty”
  • Page 4, lines 142, 146: we changed “physical activity” to “physical activity (PA)” and then we used this shortcut (PA) in all the following text.
  • Page 5, lines 194-198: we used full alignments in the text.